**communications** engineering

# A physics-informed deep learning liquid crystal camera with data-driven diffractive guidance
Jiashuo Shi[1,2], Taige Liu[1,2], Liang Zhou[1,2], Pei Yan[1,2,3], Zhe Wang[1,2] & Xinyu Zhang [1,2] ✉

Whether in the realms of computer vision, robotics, or environmental monitoring, the ability to monitor and follow specific targets amidst intricate surroundings is essential for numerous applications. However, achieving rapid and efficient target tracking remains a challenge. Here we propose an optical implementation for rapid tracking with negligible digital post-processing, leveraging an all-optical information processing. This work combines a diffractive-based optical nerual network with a layered liquid crystal electrical addressing architecture, synergizing the parallel processing capabilities inherent in light propagation with liquid crystal dynamic adaptation mechanism. Through a one-time effort training, the trained network enable accurate prediction of the desired arrangement of liquid crystal molecules as confirmed through numerical blind testing. Then we establish an experimental camera architecture that synergistically combines an electrically-tuned functioned liquid crystal layer with materialized optical neural network. With integrating the architecture into optical imaging path of a detector plane, this optical computing camera offers a data-driven diffractive guidance, enabling the identification of target within complex backgrounds, highlighting its high-level vision task implementation and problem-solving capabilities.

Tracking targets of interest (ToI) and capturing their detailed features have been longstanding challenges across various fields[1–21], including action recognition[2,4,5,14,15], biomedical optics[7,8,20,21], automatic driving[1,9], astrometry[3,10], and remote sensing[6,11,12]. The ability to efficiently track and obtain clear images of ToI using detector systems is paramount in these domains. Many image processing techniques have been developed to address essential issues such as feature extraction, filtering, and segmentation. Typically, electronic-based algorithms process both the raw image data and prior statistical information of ToI, guiding optical correction.

A classic strategy involves designing handcrafted features followed by shallow optimized architectures to extract target characteristics from input information, thereby facilitating detection[22]. However, routine performance can easily be hindered by the construction of complex ensembles that involve multiple similar local structures and high-level confusions. With the rapid development of deep learning, learning-based frameworks have emerged as a powerful tool for establishing black box mappings between ToI features and labels through data-driven massive parallel training[6,9,10,12,14].

This mapping process guides optical correction, enabling accurate detection. Despite this advancement, traditional tracking methods relying on computationally intensive algorithms are ill-suited for real-time and large-scale deployment. Although some efforts have been made to speed up the inference by employing lightweight neural network models[23,24], these frameworks rely on electrical processes such as convolution and recurrent operations, which are still computationally intensive. Different from the above digital operations, optical processes can provide ultra-fast inference speed. However, it is generally overlooked how to track ToI via an all-optical architecture.

Some recent advancements have verified that it is feasible to design optical computing architectures with distinct characteristics such as low latency, power efficiency, and parallel computing capabilities. These architectures assist or enhance machine learning hardware design[25–32], including silicon-based Mech-Zender programmable nano-photon processing[25,26] and deep spatial diffraction computing learning-based frameworks[27]. Researchers have actively explored interdisciplinary fusion, such as spatial mode sorting[33], binary classification[34], and high-dynamic

[1]National Key Laboratory of Science and Technology on Multi-spectral Information Processing, Huazhong University of Science and Technology, Wuhan 430074, China. [2]School of Artificial Intelligence and Automation, Huazhong University of Science and Technology, Wuhan 430074, China. [3]School of Computer Science and Engineering, Nanyang Technological University, Singapore 628798, Singapore. ✉e-mail: x_yzhang@hust.edu.cn

encoded mask imaging[35], to harness potential benefits. Nonetheless, implementing these techniques in high-level vision tasks remains a challenge.

This paper presents a method for tracking ToIs with negligible digital post-processing. Our approach combines a lightwave-based deep diffractive prediction with a layered liquid crystal (LC) electrical addressing architecture, offering a promising alternative that combines the parallel processing of light propagation with the robust fitting and generalization capabilities of deep neural networks. The inspiration for approach stems from the ability of deep learning architectures to optimize the weights of multiple hidden layers, which is analogous to optimizing micro-nano structures in transparent optical media.

An LC-based camera is assembled to validate the approach, comprising a functioned LC layer, a primary lens, and a detection plane. A dataset of 1613 sample scenes is collected for both training and blind testing of a 3-layer transmission phase surface using an electronic computer, where the training dataset consisted of 1452 samples, the testing dataset consisted of 161 samples. Both single-channel training (SCT) and multi-channel training (MCT) methods are employed in a gradient backpropagation optimization process to evaluate the potential impact of the proposed design. In numerical blind testing, a trained 3-layer phase surface achieves over 92.5% prediction accuracy in switching the mode of the functioned LC layer for ToI tracking, even in the presence of incomplete or defocused targets. Experimental fabrication of diffractive phase surfaces is carried out using

mask-less grayscale exposure, followed by integration into an electrically tunable LC layer. By inserting this integration into the camera's optical path, ToI hidden in a complex background can be electrically addressed, resulting in precise imaging on the detection plane. With introducing noise-injection during online training and applying physical compensation, experimental blind prediction accuracies of 61.4% are achieved, where higher precision processing and integrated molding will greatly improve its accuracy.

Our approach exhibits substantial potential for various applications in surveillance, robotics, and biomedicine fields. By designing data-driven diffractive guidance based on optical neural network, avenues emerge for developing all-optical learning-based processors tailored to high-level vision tasks. These systems demonstrate remarkable processing speeds and boast exceptional energy efficiency, thereby revolutionizing the analysis of extensive datasets. Moreover, they enable applications that are previously unattainable using conventional digital methodologies.

## Methods
### Data-driven diffractive guidance principle
An overview of a proposed data-driven ToI tracking architecture is illustrated in Fig. 1. Scenes captured by the designed LC-based camera are utilized for training a 3-layer optical neural network in Fig. 1a. This training process establishes a fundamental mapping between the input scenes containing the hidden ToI and an appropriate electrical reconfiguration of the LC molecules. An error backpropagation process is employed based on a

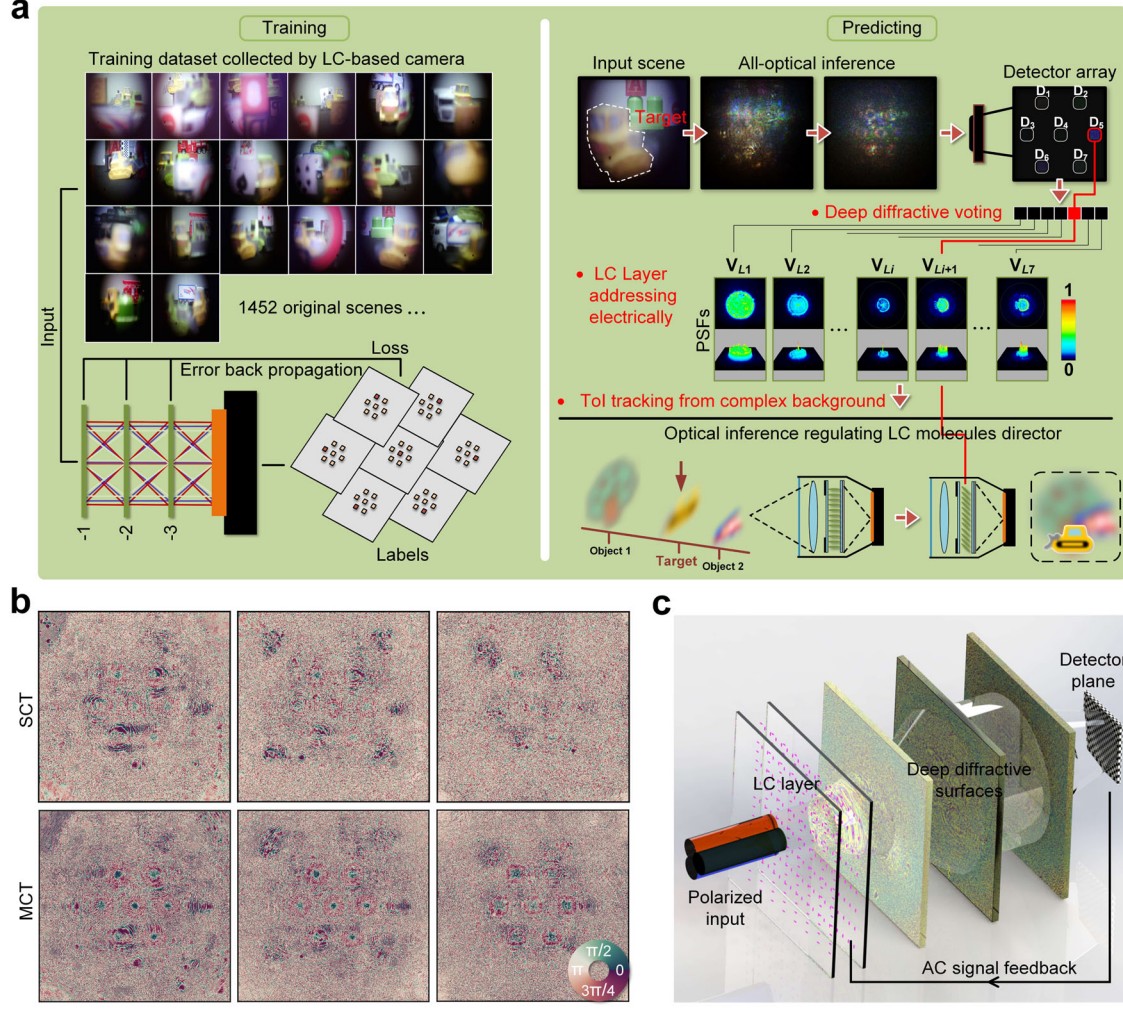

**Fig. 1 | Overview of the physics-informed deep learning liquid crystal (LC) camera architecture. a** Training and predicting principle of the proposed target tracking model. **b** Trained phase distributions of optical neural networks using 1452 original scenes. **c** Three-dimensional construction of optical computing with alternating current (AC) signal feedback for target tracking.

gradient descent strategy to switch the physical state of the LC layer and capture the desired ToI. The optimized deep diffractive phase surfaces are then used to enable all-optical inference through layer-to-layer phase modulation[27], which is a collaboration by multilayer diffraction to vote for the best answer, so we call it deep diffractive voting. This process predicts the optimal reorientation of the LC molecules, facilitating the efficient manipulation of target lightwaves to track the intended ToI.

To create the desired initial alignment of liquid crystal directors, we use a configuration where a circular-patterned aluminum electrode and a planar indium tin oxide (ITO) electrode are sandwiched together. In detail, it refers to the establishment of a nematic alignment of rod-shaped liquid crystal molecules within the liquid crystal layer. This initial alignment is achieved through the sandwich-like structure, creating either a nematic orientation of the liquid crystal molecules. This initial arrangement is crucial as it directly influences the response of liquid crystal molecules to external factors like electric fields, which, in turn, impacts the functionality and performance of the LC device. The LC materials are aligned and anchored on one smooth endface of each electrode. This setup allows us to produce modulated lightwaves that reflect the distribution of LC molecules within the LC layer. 1613 multi-object scenes are meticulously arranged by varying backgrounds, introducing jamming objects and adjusting the ToI location. These scenes are captured directly by the constructed LC-based camera, serving as the training and testing dataset. The functioned LC medium is positioned physically between the primary lens and the imaging plane of the camera's attached detector array, adhering to the aforementioned configuration.

To enhance the robustness of the training model, deliberate efforts are made to include challenging scene samples. These samples encompass a wide range of occlusion and defocusing scenarios. This study chooses the yellow engineering cars as ToIs for numerical testing and experimental processing. In extended applications, a broader training set can be provided to achieve more kinds of target recognition and tracking. It serves as the primary object in the collected samples, offering necessary spatial features and detailed textures to facilitate the learning process of the data-driven model. Unfamiliar objects and unknown background elements are also incorporated to enhance the reliability of blind testing performances. Initial sampling is performed to establish seven discrete states (i.e., the seven AC signal levels) that advanced the LC molecule rearrangement, corresponding to seven equivalent phase distributions of the LC layer, enabling variable modulation of the incident beam. In the training stage, light-intensity distributions labeled according to the most concentrated energy points, e.g., the $D_5$ in Fig. 1a, representing the LC molecule arrangement modes.

Different LC molecule arrangement could be switched by manipulating the spatial electric field in the LC layer. AC signal voltages, namely $V_{L1}$ to $V_{L7}$, are applied to control the wavefront modes characterized by point spread functions (PSFs). As indicated in the sub-figure, these PSFs are closely associated with the seven half-spherical electric fields stimulated in the LC micro-cavity. Furthermore, the position of the hidden ToI within a complex background is correlated with a specific state of the LC layer through manual calibration of labels in the training set. Each wavefront modulation corresponded to a distinct state of the functioned LC layer, aligning with the longitudinal locations of the capturing operation for the LC-based camera, which in turn correlated to ToI position in the input scenes. The collected and calibrated dataset is used for optimal phase distribution in a diffraction propagation paradigm. A multi-channel training (MCT) approach based on broad-spectrum ensemble learning[28] is performed in addition to a single-channel training (SCT) approach. Throughout the training process, both error backpropagation using a mean absolute error loss function and gradient descent in a deep learning framework is employed to update the phase distribution of these diffractive phase surfaces.

After a one-time training stage, the optimized phase surfaces are presented in Fig. 1b. Figure 1c depicts a three-dimensional schematic diagram of the proposed data-driven diffractive guidance module. This module consists of a tightly integrated functional LC layer with homogeneous alignment and the trained diffractive layers, forming a layered visible

transparent medium with unique micro-nanostructures that alter the direction of energy flux in propagating lightwaves. It is important to emphasize that, although the LC layers and diffractive surfaces are integrated, the primary optical computation and the forecasting of optimal reorientation are achieved via the diffractive phase surfaces. The LC layers serve as integral components subject to precise electrical control for the refinement of the wavefront. A polarized input enables the LC layer to have a higher modulation efficiency. By adjusting the parameters of these micro-nanostructures, the distribution of the transmitted beam can be manipulated, akin to optimizing the weights of hidden layers in a neural network. After a negligible signal extraction, the output light field utilizes a simple AC feedback loop to guide the mode switching of LC layer. Compared to an all-optical realization of a conventional neural network configuration, the diffractive-based module offers a more cost-effective and simplified optical architecture. Optical interference and diffraction phenomena enable multiple computations to occur simultaneously, which is particularly advantageous for deep learning tasks. The diffractive prediction module's depth allows for the learning of increasingly intricate optical patterns. By training a deep network, we empower the system to model and recognize intricate and nuanced ToI features.

In the proposed architecture, incident lightwaves carrying information about the ToIs location are initially focused by the primary lens, which is used for lightwave compression and imaging in ToI tracking from complex background, before entering the data-driven transparent mediums. Layer-by-layer phase-only modulation occurs as the lightwaves pass through the structured optical plane. This process yields diffractive voting lightfields, which guide selecting an appropriate voltage level to adjust the LC molecule arrangement for ToI tracking. By designing and integrating the trainable deep diffractive layers with the electrically controlled LC layer, the proposed architecture enables efficient ToI tracking by combining rapid lightwave propagation with deep networks' powerful fitting and generalization capabilities.

## Data acquisition and phase surface training

For facilitating the training of the deep diffractive phase surfaces, a dedicated LC-based camera is constructed for data collection, calibration, and continuous testing of the dataset required during the phase optimization process, as depicted in Fig. 2. The camera incorporates a functional LC layer between the primary lens and the detector, which is located at a distance of 8 mm from imaging plane, establishing an LC-based imaging system. This setup is connected to an AC generator using conductive tape. The scenes captured by the camera are divided into seven layered spatial areas, denoted by the marker set {∞, $R_1$, $R_2$, $R_3$, $R_4$, $R_5$, $R_6$, $R_7$}, based on the clarity of the images obtained under different voltage loading levels. The voltage level directly corresponds to the physical state of the functional LC layer. For instance, an AC voltage of approximately 25.4Vrms induces an appropriate LC molecule reorientation or rearrangement, resulting in a desired refractive index distribution and relatively clear imaging in the spatial region of [$R_3$,$R_4$]. The signal voltage level correspond to the electric field distribution across the LC layer. To achieve this distribution, we have designed a sandwich structure for the LC layer, along with patterned electrodes. By adjusting the voltage applied to the electrodes, we can effectively switch and control the electric field distribution within the LC layer. The specific details of the electric field distribution switching are visually represented in the Supplementary Note 3. A complementary metal oxide semiconductor (CMOS) camera fixed with a 3D-printed UV photosensitive resin is placed at a consistent distance of approximately 8.4 mm from the functional LC layer. This ensures clear detection from $R_1$ to infinity in the initial state. Please note that we only performed spatial segmentation in the depth direction to track the ToIs region to verify the rationality of the proposed data-driven diffractive guidance method. In fact, the x-y plane segmentation can also be established according to the large-scale dataset as well as the PSFs spatial variation about the LC layer.

Each captured scene is divided into seven parts based on the ToIs position, forming training pairs with corresponding calibration labels in an

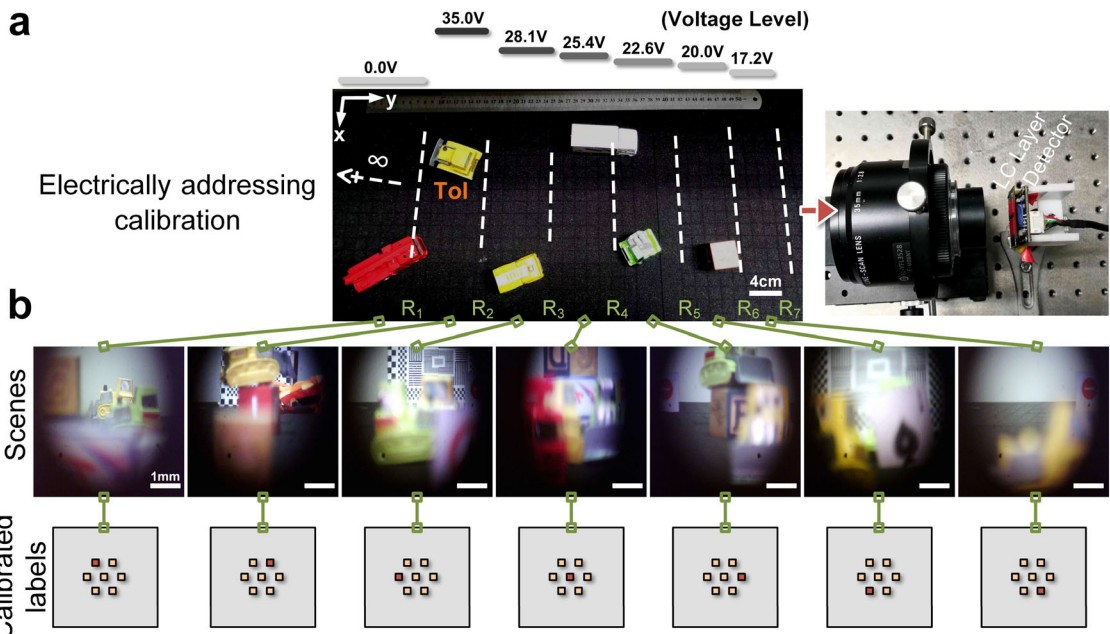

**Fig. 2 | Acquisition of training data and liquid crystal (LC) layer electrical calibration for target addressing. a** A liquid crystal camera collects training scenes by adjusting the target of interest (ToI) position and background. **b** Based on the location of the target, several labels are assigned to each optical image respectively.

electronic computer. Approximately 200 digital images are maintained for each component to ensure a uniform training process. In total, 1613 digital images are utilized, with 1452 images allocated to the training set and 161 to the testing set. These images are employed for online optimization and assessment of the 3-layer deep diffractive phase surface. After preliminary evaluation, 1452 samples are sufficient to avoid overfitting of the training model. To minimize oscillation during gradient backpropagation optimization, the adaptive moment estimation (Adam) optimizer[36] is utilized with a learning rate of $10^{-3}$, incorporating momentum estimation and dynamic learning rate decay. The trainable deep diffractive propagation model is built using Python 3.75 and the TensorFlow deep-learning framework (version r2.40).

SCT and MCT methods are employed in the training stage to form a controlled experiment. Specifically, three wavelengths of [460 nm, 550 nm, 640 nm] are selected, corresponding to the peak responses of the actual CMOS color response curves. These wavelengths exhibit a relatively high Quantum Efficiency (Q-E) conversion efficiency of [79.3%, 92.5%, 90.1%] for ensemble learning in MCT[28]. SCT is only using incident light at 550 nm. The beam diffraction propagation model and the main parameters of the diffractive surfaces remain consistent across both methods.

In the numerical diffraction model, we adopt a Fourier-based representation of the Rayleigh-Sommerfeld scalar diffraction formula to approximate the free-space propagation of lightwaves in the 3-layer phase-only modulation process. The layered neural network is configured with dimensions of $800 \times 800$, ensuring a sampling interval of approximately 5 μm and a layer spacing of roughly 2 cm. This parameter setup considers the maximal half-cone diffraction angle of the fully connected model formed by the secondary lightwaves, ensuring optimal phase utilization efficiency and robust model generalization capability. On the other hand, higher resolution will reduce experimental accuracy due to the exponential increase in processing difficulty. Additionally, edge padding is implemented during the diffractive transformation and convolution operations to minimize edge calculation errors.

To emulate realistic conditions, we have incorporated misalignment errors ranging from 0.5 to 2% of side length and height errors from 5 to 8% of phase steps. These errors simulate the precision limitations of typical processing and mask-less grayscale exposure. Additionally, Gaussian and Poisson noise with hyperparameters $\sigma$ and $\lambda$ set to 0.001 has also been introduced to simulate dark current noise. The model will have better robust

performance under various conditions by utilizing noise-injection in training process. Furthermore, in the absence of featured information about the ToI for other wavelengths, SCT is more prone to prediction errors than MCT. We conduct a preliminary verification for the above deduction by testing set. Under different layer configurations (2-layer, 3-layer, and 4-layer), MCT achieved average accuracies of 81.2%, 92.5%, and 87.9%, respectively. Similarly, SCT achieved average accuracies of 71.6%, 76.5%, and 72.3% with five rounds of retraining from the same initial state. Based on these results, we have selected the 3-layer diffractive surface as the focus of our study.

## Results and discussion
### Numerical assessment of ToI tracking
In this section, we meticulously assess the inference operation of the deep diffractive phase surfaces across a divided range of ToI locations. The optimized phase surfaces are obtained using both SCT and MCT. Their performance is evaluated in detail, as depicted in Fig. 3. We provide a comprehensive overview of the evaluation process through numerical optical inference results, showcased in Fig. 3a. The first column is the image on the CMOS with just the lens and the LC layer in place where columns 2-4 are numerical results of what the CMOS will see if the trained deep diffractive layers are in place. We draw conclusions in column 5 based on the energy distribution observed in the inference results of column 4.

Figure 3b presents a numerical representation of the spatial distributions stimulated by a specific signal voltage applied across the LC layer alongside several typical LC molecule arrangement patterns obtained through deep diffractive predictions. It refers to different patterns in which the LC molecules are distributed within the LC layer. Each of these patterns corresponds to a specific working focal length or target capture depth within the LC layer. The proposed optical system employs these various distribution modes to selectively achieve different working focal lengths or depths of field, enabling the tracking of ToI. These predictions correspond to input scenes where the ToI is located in different regions labeled as $[R_6, R_7]$, $[R_3, R_4]$, $[R_2, R_3]$, and $[R_1, R_2]$. To assess the blind testing performance, we provide main simulation results from various scenario samples with multiple disturbances. These results showcase the identification of specific ToI, denoted by white lines in the input samples. The input scenes are then transformed into characterized light-intensity distributions containing essential information about the ToI position for decision-making purposes.

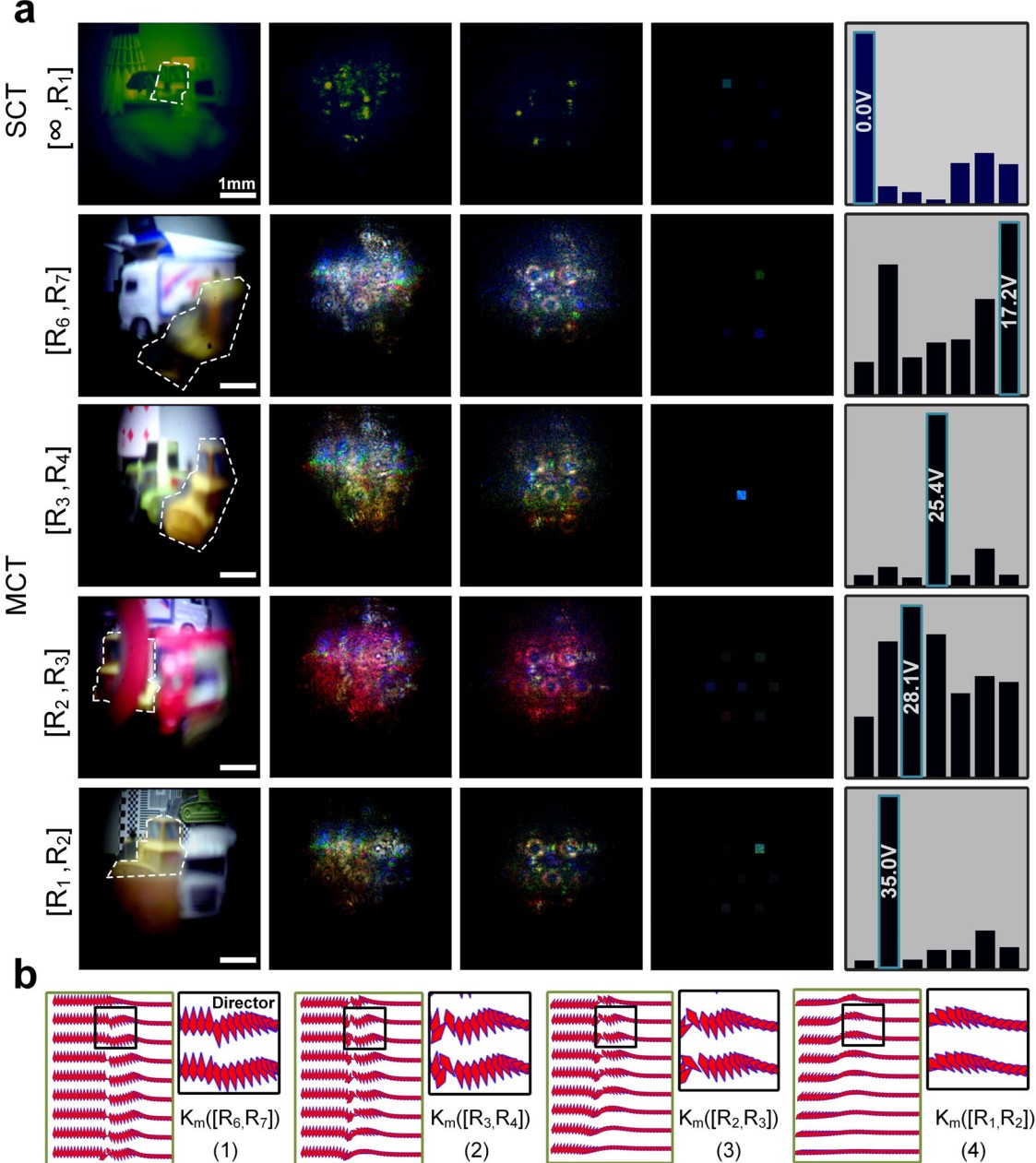

**Fig. 3 | Evaluation of various scenario samples corresponding to diverse target of interest positions for assessing the trained diffractive phase morphology simulation's inference performance. a** Input optical images modulate and transform layer by layer into an output lightfield, predicting a corresponding liquid crystal molecule arrangement based on single-channel training (SCT) and multi-channel training (MCT). **b** Typical liquid crystal molecule distributing patterns across various molecule arrangement modes.

This transformation is achieved through lightwave propagation based on Rayleigh-Sommerfeld diffraction and phase modulation.

Careful analysis is conducted on seven detecting sub-regions, corresponding to the output light intensity distribution over each viewing surface, to shape a normalized energy distribution. This aids in predicting a suitable signal voltage that leads to a layered LC molecule rearrangement aligned with the desired cascaded deep diffractive surface. The most energetically favorable region among all areas provides a set of predicted signal voltages to be applied across the functioned LC layer. Subsequently, these voltages stimulate LC molecules to undergo orderly reorientation based on the electric field simulated between electrode plates, using a set of appropriate root-mean-square (rms) voltages. These voltages are indicated as $0Vrms@[\infty,R_1]$, $17.2Vrms@[R_6,R_7]$, $25.4Vrms@[R_3,R_4]$, $28.1Vrms@[R_2,R_3]$, and $35Vrms@[R_1,R_2]$.

A typical process can be summarized as follows: First, a lightfield containing the ToI in the lower right region $[R_6,R_7]$ undergoes conversion into a predicted light intensity distribution. This conversion is achieved through cascaded and controlled diffraction phase modulation, employing a standard angular spectrum model. The energy distribution formed from this process allows for extracting a predicted signal voltage of 17.2Vrms. Even when the ToI is obscured and segmented by a complex background, a correct distribution of LC molecules can still be shaped through data-driven robust predictions obtained from the deep diffractive process. This is illustrated in the second subfigure of the fourth column in Fig. 3b. The numerical spatial electric field stimulated by the signal voltage of 17.2Vrms results in the formation of typical distribution based on the deep diffractive prediction. Here, "$K_m$" represents the trained mapping relationship between the ToI position and the rearrangement of layered LC molecules, which is

closely linked to the stimulated electric field morphology within the LC layer. The parameters $K_m([R_6,R_7])$, $K_m([R_3,R_4])$, $K_m([R_2,R_3])$, and $K_m([R_1,R_2])$ specifically indicate the corresponding LC molecule distribution patterns determined by the electric field profile. Thumbnail images further illustrate more detailed deflection angles of the LC director, where different director arrangements correspond to other modes of LC reorientation, displaying slight discontinuity.

In summary, the LC-based camera efficiently tracks the position of the ToI in real time by generating, switching, and adjusting the phase distribution through the functioned LC layer based on adaptive predictions of LC molecule rearrangement morphology from a set of learning-based deep diffractive phase surfaces. Both training strategies mentioned above numerical result in accurate predictions for the actual position of the ToI, enabling the imaging system to track it, as depicted in Fig. 3.

## Fabrication and characterization

For experimental verification viability of the proposed data-driven diffractive guidance, several diffractive elements, including a trained 3-layer SCT-based phase surface and a 3-layer MCT-based phase surface, are fabricated by a mask-less grayscale exposure. A total of 16 phase levels are achieved finally in a single UV-exposure operation, as shown in Fig. 4.

Elaborately, a 0.5 mm-thick fused silica wafer is utilized as a substrate in our fabrication process. The substrates are first ultrasonically cleaned by deionized water. And next, a film of positive photoresist (AZ1518, Micro Chemicals) is spin-coated over the surface of the substrate utilizing a spin processor at 3000 rpm for 40 s, then heated at 115 °C for 90 s. The refractive index of the formed AZ1518 film is finely measured using a Spectroscopic Ellipsometry Analyzer (Semilab SE-2000). The actual refractive index data of 1.7302@460 nm, 1.7091@550 nm, 1.6989@632.8 nm, 1.6983@640 nm, and 1.6824@980 nm are acquired. All trained heightmaps are then converted into a set of 800×800 BMP grayscale images having a 5 μm pixel spacing, which are then imported into a mask-less lithography machine (Heidelberg DWL66 + ), as shown in Fig. 4a. Finally, the designed patterns are transferred from the initial photomask into the photoresist under a single UV exposure with a central wavelength of 375 nm.

The chemical properties of the photoresist in the exposed region will undergo a remarkable change during continuously rendering it removable in the base developer (MF-319) for 35 seconds. The total fabrication depth with 16 phase levels is ~2000 nm. Accordingly, each phase level in our design has a depth of ~125 nm. Our final depth error is confined to an accuracy range of ±15 nm. Here we present microscope images of the fabricated phase surfaces, as shown in Fig. 4b, d. The detailed functional micro-nano-structures are successfully acquired by comparing the numerical phase distribution shown in Fig. 4c and the actual diffractive surface shown in Fig. 4d according to a common optical microscope characterization.

These fabricated diffractive phase surfaces, featuring a square aperture of approximately 4 mm side length, are mounted onto a metal groove, as depicted in Fig. 4h. The metal surface is treated with matte spray paint to mitigate boundary reflections, creating an approximate boundary absorption condition. The point spread function of the deep diffractive phase surfaces corresponding to SCT and MCT is determined by measuring the optical response of a collimated white beam. Figure 4e, f displays the obtained point spread functions, with a hexagon marking the position of seven feature detection points. An experimental architecture for ToI tracking is illustrated in Fig. 4g. It comprises several vital modules, including the functioned LC layer, the deep diffractive phase surfaces, and the imaging and detector components. The diffraction channel is left uncovered to provide a clear view of the internal structures. Still, a metal absorption boundary is wrapped around it in experiments to reduce stray beams during propagation. In practice, a spectroscope is used to separate the incident beam for optical calculation and imaging respectively, thus achieving all-optical linkage.

Figure 5 shows the fabrication and characterization of the functioned LC layer. In Fig. 5a, the fabricated LC layer is visually presented. A polyimide layer is spin-coated onto the endfaces of a circular-patterned aluminum

electrode and a planar ITO electrode to achieve the desired performance. The polyimide layer is prebaked at 80 °C for 10 min, then cured at 230 °C for 30 min. These cured polyimide layers serve as initial alignment and anchoring coatings for the LC material (E44, $n_e$ = 1.7904 and $n_o$ = 1.5277), which are rubbed anti-parallel to achieve a homogeneous alignment of LC molecules[37]. The adhesive mixed with 20 μm diameter glass microspheres are also spread on the sides of the glass substrate. These microspheres act as spacers, separating the two substrates and supporting the intended microcavity shape. The fabricated LC layers are loaded with an AC signal using conductive tapes pre-connected to their patterned aluminum and ITO electrodes, where the aperture of the LC layer is the same as in Fig. 1, to ensure the training data is more in line with the actual scene. In the Supplementary Note 4, we include a schematic diagram of the aluminum electrode, along with its dimensions, to enhance the visual representation and comprehensibility of this element in the experimental setup. The alignment and anchoring of the liquid crystal molecules in this study are achieved using a planar alignment. This alignment is realized by adding an anisotropic alignment layer between the liquid crystal and the electrodes, which takes advantage of the anchoring effect on the liquid crystal's surface. The material used for the alignment layer is polyimide, and the method involved frictional rubbing of the polyimide coating with cotton cloth. Through this process, microgrooves are formed on the surface of the polyimide coating, enabling the liquid crystal's director vectors to align parallel to the direction of these grooves. Additionally, during the frictional rubbing process, the polyimide material generated polymer chains. The intermolecular forces between the liquid crystal and the polymer chains contributed to an increased anchoring energy. This planar alignment method with the polyimide alignment layer allowed the liquid crystal molecules to align in parallel, in accordance with the predefined direction, and enabled the desired phase modulation properties and functionality of the liquid crystal layer.

The optical response of the functioned LC layer, including the point spread functions (PSFs), is measured using a collimated white beam, a linear polarizer, and a ×10 microscope objective. Figure 5b illustrates a mild compressed lightfield mode of LC layer using an AC signal of 5 KHz@20Vrms. By adjusting the output of the AC generator, the focal length corresponding to the incident beam compression mode is varied at different driving voltages, as depicted in Fig. 5c. LC layer performs an adjustable focal length that can vary within the range of 4.3 to 8.1 mm. The thumbnails in the figure display specific LC modes characterized by their point source response, corresponding to applied signal voltages of 13Vrms, 15.2Vrms, 22.6Vrms, 32.9Vrms, and 35Vrms. These thumbnails provide a detailed view of the LC modes shaped by incident wavefront modulation. Hence, the trained deep diffractive phase surfaces can be utilized to switch the mode of the functioned LC layer. This allows for efficient capturing of the ToI by providing a modulated light intensity distribution based on a predicted AC driving signal voltage.

## Experimental evaluation of ToI tracking

In this section, we present the experimental all-optical predictive performances of the proposed ToI tracking method, which employs an optical layout consisting of a functioned LC layer, 3-layer deep diffractive phase surfaces, imaging and detecting components, as depicted in Fig. 4g. The visible scene information is initially captured by a primary lens and recorded by a high-resolution CMOS camera.

In Fig. 6, we present experimental confirmation of learning-based diffractive guidance for LC molecule alignment, leading to a comprehensive rendering of the ToI. Figure 6a–h showcases the input scenes and their corresponding output distributions. Figure 6i visually represents the LC mode switching. Finally, the resulting confusion matrices are depicted in Fig. 6j. We employ two typical scenes (sub-figures (a) and (b)) as untrained samples. The ToI is located in $[R_2, R_3]$ and $[R_6, R_7]$ regions. Additionally, several model toys, serving as distractors with similar colors or shapes, are placed around the targeted ToI to increase the task's difficulty.

Based on the proposed architecture approach, the tracking and recognition of the ToI rely on the LC layer's switching to an appropriate molecule arrangement state. The output results based on SCT and MCT, as depicted in sub-figures (c) to (e) and (d) to (f), respectively, showcase the lightwave intensity distributions. High intensity points are marked in these sub-figures to demonstrate the all-optical inference accuracy. These results guide the operation of mode switching of the LC layer. The energy distribution percentage (EDP), which is defined as the energy percentage of the high intensity point in the total energy of detector array, of concentration points in the output intensity distribution, shown in sub-figures (c) and (e),

is presented in the thumbnails of sub-figures (g) and (h). A higher EDP generally ensures a reliable optical prediction result while remaining distinguishable from dark current noise during the photoelectronic detection step. The primary reason for the visual contrast between scenes in Fig. 3a and Fig. 6a, b is that they represent different stages of the experimentation. Figure 3a corresponds to the numerical input, where we can control parameters for illustrative purposes. In contrast, Fig. 6a, b represents the actual experimental input. During experiments, it is often necessary to introduce a stronger background illumination to enhance the contrast of the optical network's predicted light field. This can lead to a perceived difference in the

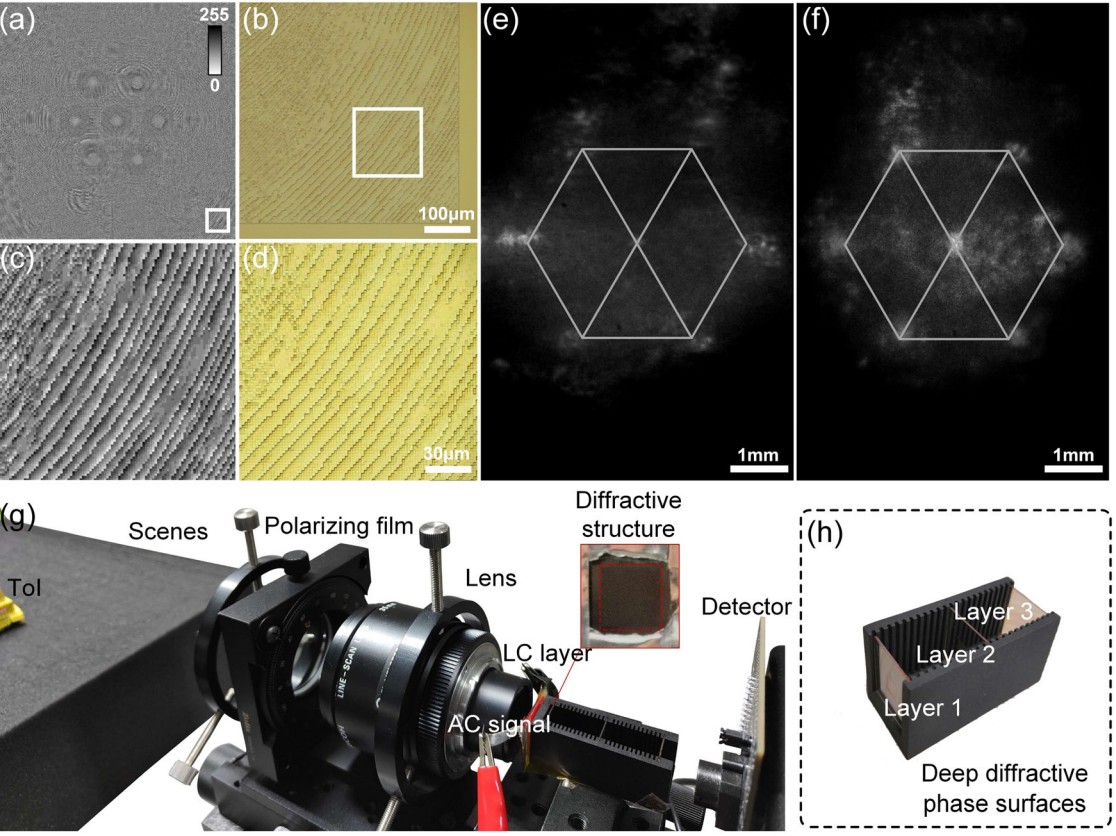

**Fig. 4 | Fabrication and characterization of the trained physics-informed phase surfaces of optical neural network. a, b** Display the phase distributions for the first layer of optical neural network. Employ a white frame to extract matching regions in (**c**) and (**d**) for scrutinizing their detailed features. The point spread functions of the

single-channel training and multi-channel training method are depicted in (**e**) and (**f**). **g** Illustrates the designed liquid crystal (LC) camera driven by alternating current (AC) signal employed for tracking the target of interest (ToI) selected, where a detailed data-driven diffractive guidance is demonstrated in (**h**).

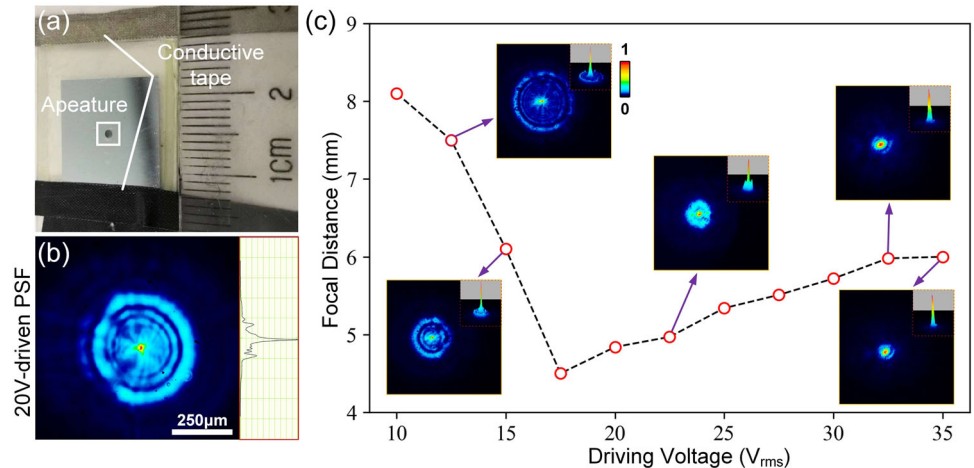

**Fig. 5 | Fabrication and characterization of the functioned liquid crystal layer. a** Depicts a custom-designed liquid crystal layer featuring a circular-patterned aluminum electrode and a planar indium tin oxide electrode. The optical response, i.e. the point spread function (PSF), of liquid crystal layer to collimated white beams with a basic set of parameters of 5 kHz frequency and 20 Vrms AC signal, is presented in (**b**). As the driving signal voltage varies, a mode switching according to an optical response of the liquid crystal layer is demonstrated in (**c**).

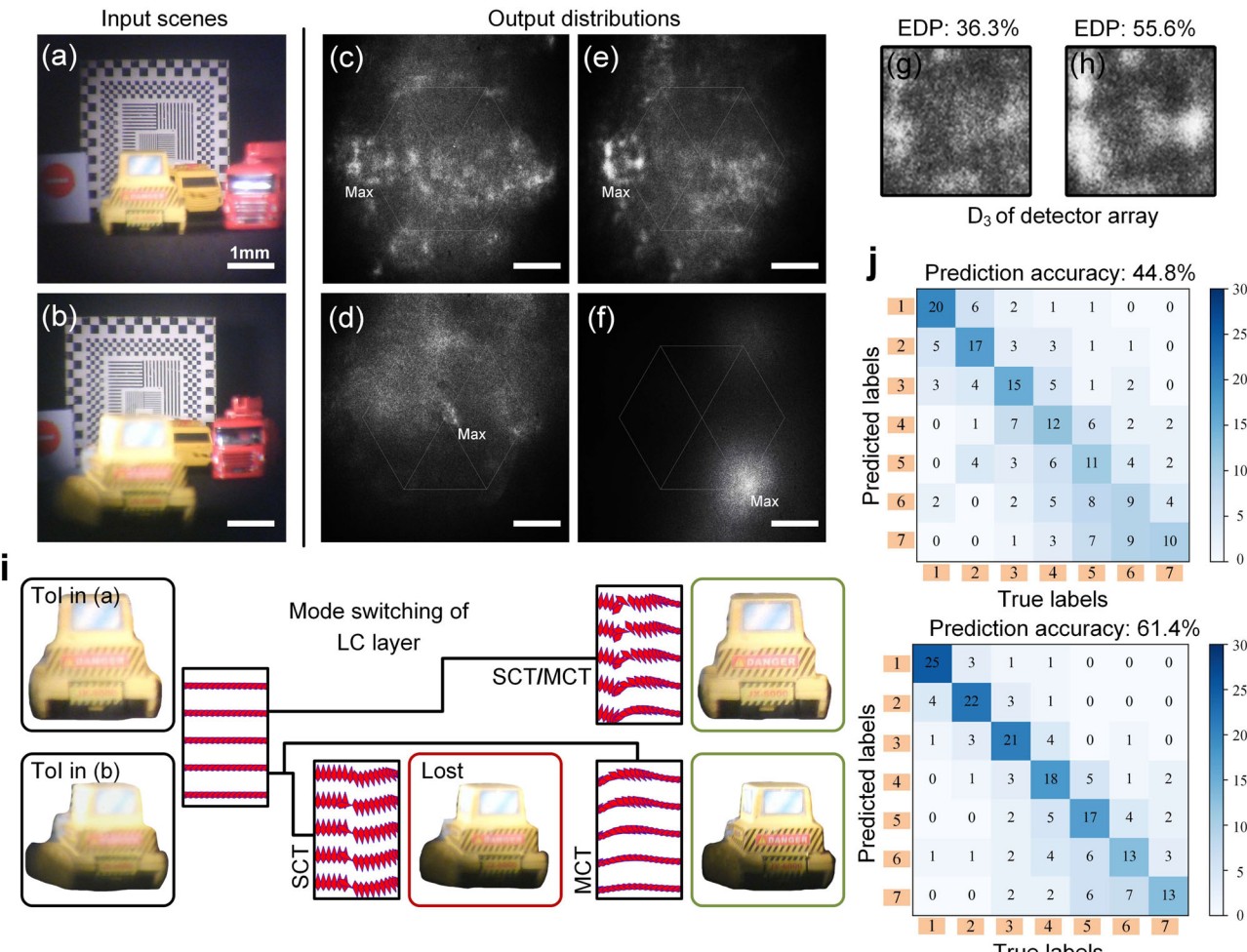

**Fig. 6 | Experimental confirmation of the data-driven diffractive guidance for shaping a desired liquid crystal (LC) molecule alignment in comprehensive rendering of the target of interest (ToI).** Testing scene shown in (**a**) and (**b**) are related to actual prediction outputs shown in (**c–f**) according to single-channel training (SCT) and multi-channel training (MCT) method, where those in (**g**) and (**h**) depict the energy distribution percentage (EDP) of the concentration points in output intensity distributions. **i** Typical depiction of the functioned liquid crystal layer switched for scenes (**a**) and (**b**). **j** Confusion matrices correspond to single-channel training and multi-channel training method.

visual representation. In the Supplementary Note 1 and Supplementary Movie 1, we adjust the intensity and angle of the background illumination to weaken this illusion.

Figure 6i comprehensively depicts the mode switching process for scenes (a) and (b). The initial alignment mode of the LC layer is guided by the diffractive operation of the 3-layer phase surface, enabling lightwave-based inference and effective adaptation of the ToI's movement. For scene (a), both SCT and MCT accurately predict the necessary LC molecule alignment to regulate the wavefront propagation towards the photosensitive plane of the LC-based camera, resulting in a comprehensive rendering of the ToI's details. In scene (b), using additional color information by MCT enhances the stability of ToI tracking performance.

Figure 6j presents two comprehensive confusion matrices, corresponding to SCT and MCT, obtained through a testing process involving 30 distinct samples in 7 detection depths. Each confusion matrix is structured as a 7 × 7 matrix, aligning with the number of categories in the classification task. The rows in the matrix represent the model's predictions, while the columns represent the true labels. Each cell in the matrix represents the relationship between the predicted outcomes and the true labels. Our findings indicate that optical predictions are more prone to errors in adjacent regions of the objective space resulting from the similar blurring of the ToI. Notably, our proposed methodologies have demonstrated experimental performance, achieving over 61.4% prediction accuracy for high-

level vision task predictions, as evidenced by the comprehensive experimental results. Furthermore, We also provide a comparative analysis of the prediction time and accuracy for proposed method and conventional digital neural network in Supplementary Note 2. The performance reported in Fig. 6j represents the accuracy achieved under specific conditions and may not necessarily reflect the performance limit of the proposed configuration. The achievable performance can be influenced by various factors, including the complexity of the ToIs, the quality of the training dataset, and the errors in physical processing. It can be expected that higher experimental accuracy can be achieved using higher precision layer arrangements, such as two-photon direct write printing. Achieving optimal results and pushing the boundaries of this configuration will require advancements in both manufacturing accuracy and the sophistication of optical propagation models. All in all, these outcomes exemplify the effectiveness of our approach in delivering robust and reliable predictions, even in the presence of complexities introduced by adjacent regions and the inherent blurring.

## Conclusions

In conclusion, we have introduced a approach for tracking Targets of Interest (ToI) with negligible digital post-processing. Our method combines a deep-learning-based diffractive prediction with a functioned LC-layer addressing architecture. This approach offers several advantages, including high-speed manipulation of target lightwave propagation and spatial

distribution and robust fitting and generalization capabilities of deep optical neural networks. The experimental validation of our proposed functional layout demonstrates its potential for rapid and intelligent capturing of real-world scenarios. The designed target detection architecture and the featured information processing approach represent a step forward in developing all-optical, learning-based target processors for high-level vision tasks. Overall, our work opens up possibilities for efficient and intelligent tracking of ToI, showcasing the potential of all-optical approaches in addressing complex vision tasks.

## Data availability

Data underlying the results presented in this paper are not publicly available but may be obtained from the authors upon reasonable request.

## Code availability

All custom code used in this work, including that used to train and test the framework, can be obtained from the following publicly accessible resource[38].

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

## Acknowledgements

This work was supported by the National Natural Science Foundation of China (61176052) and Fundamental Research Funds for the Central

Universities (HUST 2022JYCXJJ002). The authors would like to thank Wen Liu and Chenggang Zhou from the nanoscale research and fabrication center, University of Science and Technology of China, for their support in designing and prototyping diffractive optical elements and my fiancée Qing Liu for proofreading.

## Author contributions

Conceptualization: J.S.S., X.Y.Z. Methodology: J.S.S., T.G.L., L.Z., P.Y., Z.W. X.Y.Z. Investigation: J.S.S., L.Z. Visualization: J.S.S., X.Y.Z. Funding acquisition: X.Y.Z. Supervision: X.Y.Z. Writing – original draft: J.S.S., L.Z. Writing – review & editing: J.S.S., X.Y.Z.

## Competing interests

The authors declare no competing interests.
