## [Peer Review File · Communications Engineering]

Reviewers' comments:

Reviewer #1 (Remarks to the Author):

The authors propose an all-optical based configuration to perform tracking of Targets of Interest (ToI) that requires only negligible digital post-processing. The all-optical processing includes a deep learning-based diffractive prediction module including a tunable liquid crystal (LC) layer.

The presented concept is both novel, interesting and applicable. The paper is clear and well written and therefore could be published after revision. However, before being accepted for publication, I suggest the authors to address the following aspects:

- It is not clear why the proposed diffractive based module was selected. I mean what are its advantages in respect to e.g. all-optical realization of conventional neural network configuration? Also, it is not clear how the LC layer together with the diffractive prediction module allow sufficient amount of freedom and flexibility to train the all-optical configuration to be able to detect ToI with sufficient versatility as required in real case scenario.
- What is the estimated performance envelope for the sensitivity and the precision parameters for the trained processor? The authors should show the confusion matrix they get from their experiments with numbers and not with colors as appears in Fig. 6(c). It is mentioned in Fig. 6(c) that the device achieves prediction accuracy of 61.4% or 44.8%. This is quite low performance. Is this the performance limit for the proposed configuration?
- It is not clear how large was the training dataset. Also, the experimental validation is very limited/small scale and in my opinion it should be expanded to better represent functionality in real case scenario. Fig. 6 by itself is very preliminary and it is recommended to try the approach on larger scenario possibilities.
- The authors state that “The experimental validation of our proposed functional layout demonstrates its potential for rapid and intelligent capturing of real-world scenarios”. Thus, it could be good to compare the proposed performance with existing approaches. Especially to show how much processing complexity and time the proposed all-optical configuration saves in comparison to conventional digital post processing approaches.

Reviewer #2 (Remarks to the Author):

The authors describe an analog version of deep learning technique of image analysis. Comparing to its digital counterpart the analog technique offers a number of advantages well described in the paper, although requires rather lengthy fabrication of diffractive elements that will only be used for single purpose (single tracking object or a scene). The idea is new, important and timely and definitely requires careful attention. It perfectly fits the scope of the journal, would be of interest to others in the field and worth publishing.

However, the paper needs some more work. After reading it a few times and putting quite an effort to understand, I still have a few questions about the basics.

LC element.

1. Is it the same object (LC element) on Fig.1a with aperture about the lens size and Fig.5a with aperture about 1mm?
2. Its description, lines 100-104, is unclear.
3. 101 "desired initial arrangement of LC directors" which is what?
4. 102 "aligned and anchored" how? Planar? Homeotropic? And why?
5. 312 "In Fig. 5, the fabricated LC layers are optically characterized" and then all paragraph about fabrication, not characterisation.
6. 313 "circular-patterned aluminum electrode" probably aluminium. A schematic picture would be helpful here. What are the dimensions?
7. 317 "homogeneous alignment" OK. On Fig.1 it looks like homeotropic alignment.
8. 318 "microspheres is also spread over a glass substrate" everywhere? including the LC area or only on the sides? English also needs some correction here.
9. 321 Fig. 5c "Focus(mm)" do you mean focal distance or something else?
10. As I understand the LC element is a voltage tunable lens with focal length between 4 and 8 mm. Is it right?
11. Is the same LC lens is used for all the steps of the experiment?
12. 163 "LC layer between the primary lens and the imaging plane" if this is an additional tunable lens we need to know it's position in the optical system.

Diffraction elements

1. 110 Fig. 1b Phase distributions. How were they obtained?
2. 181 "These images are employed for online optimization and assessment of the 3-layer deep diffractive phase surface." How exactly this is done?
3. Would be good to show a comparison to digital neural network arrangement. In particular, what is equivalent of the nonlinearity (the scaling function) in each layer, which is the key element of a digital neural network layer.

Calibrated labels

1. 205 "several labels are assigned to each input sample respectively" how?
2. 225 "The first column is the image..." "columns 1-3 are numerical results" probably columns 2-4?
3. 233 "typical LC molecule distributing pattern across various LC molecule arrangement modes." Unclear what it is and how this is used.

Reviewer #3 (Remarks to the Author):

In this paper, the authors presented an LCD camera with diffractive surface-based guidance to track a target in the depth direction. They employed a deep diffractive system to predict the LCD voltage, subsequently adjusting the PSF/focal plane of the system. However, the manuscript lacks some pivotal

convincing results and I cannot recommend it for publication.

1. The author showed the proposed architecture in Fig. 1, yet it is confusing. From Fig. 1a, it seems that the authors utilize diffractive phase surfaces to predict the optimal reorientation of the LC molecules, and tracked the target from the complex background using LC molecules only. Conversely, Figure 1c suggests that the LC layers are integrated with the diffractive surfaces, implying that both the LC layers and diffractive surfaces collaboratively track the target. The authors must elucidate their methodology more distinctly.
2. Following the previous question, the authors mentioned a primary lens was employed. Is it used in “optical inference” or “ToI tracking from complex background”?
3. In the paper, the authors stated that they “only performed spatial segmentation in the depth direction to track the ToIs region”. What’s the axial resolution achieved by the deep diffractive voting process? A characterization of how performance is impacted by the position of ToI within the FOV (on x-y plane) is needed.
4. Regarding each signal voltage level, is it a singular value or an electric field distribution across the LCD layer? If it's a distribution, how do the authors determine it?
5. Throughout the manuscript, the authors emphasize tracking the target of interest. However, both the simulation and experimental results primarily depict the prediction of LC molecule arrangements via deep diffractive layers. It would be beneficial for the authors to provide visual evidence of the actual target tracking.
6. I observed marked differences between the input scenes in Fig. 3a and Fig. 6a. The field of view (FOV) in Figure 3a appears more constricted, with objects seeming blurry. In contrast, the fov in Figure 6a is considerably broader, presenting objects with clarity. Could the authors clarify the reasons behind this discrepancy?
7. “1-3 are numerical results ...” in line 226 should be “2-4 are numerical results...”

Reviewer #1 (Remarks to the Author):

The authors propose an all-optical based configuration to perform tracking of Targets of Interest (ToI) that requires only negligible digital post-processing. The all-optical processing includes a deep learning-based diffractive prediction module including a tunable liquid crystal (LC) layer.

The presented concept is both novel, interesting and applicable. The paper is clear and well written and therefore could be published after revision. However, before being accepted for publication, I suggest the authors to address the following aspects:

-- We sincerely thank the reviewer for his/her constructive feedback and positive evaluations.

(1) It is not clear why the proposed diffractive based module was selected. I mean what is its advantages in respect to e.g. all-optical realization of conventional neural network configuration? Also, it is not clear how the LC layer together with the diffractive prediction module allow sufficient amount of freedom and flexibility to train the all-optical configuration to be able to detect ToI with sufficient versatility as required in real case scenario.

-- This is an important point that the referee is pointing to. To address this comment, we modified our discussion section with the following sentences:

“Compared to an all-optical realization of a conventional neural network configuration, the diffractive-based module offers a more cost-effective and simplified optical architecture. Optical interference and diffraction phenomena enable multiple computations to occur simultaneously, which is particularly advantageous for deep learning tasks.”

“The diffractive prediction module's depth allows for the learning of increasingly intricate optical patterns. By training a deep network, we empower the system to model and recognize intricate and nuanced ToI features.”

The choice of a diffractive-based module in our work stems from several considerations. First, compared to an all-optical realization of a conventional neural network configuration, the diffractive-based module offers a more cost-effective and simplified optical architecture. Traditional all-optical neural networks often require complex optical elements and extensive optical paths, leading to higher costs and increased complexity in practical implementations. In contrast, our approach simplifies the optical setup by leveraging diffraction, making it more accessible and affordable. Second, the diffractive-based module provides inherent advantages in terms of parallel processing and speed. Optical interference and diffraction phenomena enable multiple computations to occur simultaneously, which is particularly advantageous for deep learning tasks, especially when dealing with large-scale datasets. The parallelism of our approach can significantly enhance processing efficiency.

For freedom and flexibility, the LC layer provides programmable phase modulation, allowing precise control over the phase of transmitted light. As we deepen the stack of diffractive layers within the module, we introduce additional degrees of freedom, which enable more complex phase profiles. This results in the ability to engineer a wider variety of optical transformations, further enhancing the system's adaptability. Secondly, the diffractive prediction module's depth allows for the learning of increasingly intricate optical patterns. By training a deep network, we empower the system to model and recognize intricate and nuanced ToI features. The added depth in the network

enhances its capacity to handle a broader spectrum of ToI characteristics in real-world scenarios. In summary, the combination of the LC layer and the diffractive prediction module provides the flexibility needed to detect diverse ToI types, and the increasing depth of the diffractive layers augments the system's freedom to adapt to various real-case scenarios.

(2) What is the estimated performance envelop for the sensitivity and the precision parameters for the trained processor? The authors should show the confusion matrix they get from their experiments with numbers and not with colors as appears in Fig. 6(c). It is mentioned in Fig. 6(c) that the device achieves prediction accuracy of 61.4% or 44.8%. This is quite low performance. Is this the performance limit for the proposed configuration?

-- We thank the reviewer for this valuable comment. To address this comment of the referee, in our revised manuscript:

Fig. x1 Prediction accuracy of three methods

“The performance reported in Fig. 6(c) represents the accuracy achieved under specific conditions and may not necessarily reflect the performance limit of the proposed configuration. The achievable performance can be influenced by various factors, including the complexity of the ToIs, the quality of the training dataset, and the errors in physical processing.”

Fig. x2 Confusion matrix and prediction accuracy

In Fig. x1, we give the estimated performance envelop of CNN(3-layer), FCNN(3-layer), and the proposed method. Through the tests that are concentrated in 6 different verification sets, we give the variance of prediction accuracy for three methods.

We appreciate your feedback regarding the presentation of the confusion matrix. In the revised manuscript, we will provide a numerical representation of the confusion matrix as shown in Fig. x2, which will allow for a more detailed and transparent evaluation of the model's performance. This will include the numbers of true positives, true negatives, false positives, and false negatives to provide a clear and precise assessment of the model's classification accuracy.

The performance reported in Fig. 6(c) represents the accuracy achieved under specific conditions and may not necessarily reflect the performance limit of the proposed configuration. The achievable performance can be influenced by various factors, including the complexity of the ToIs, the quality of the training dataset, and the errors in physical processing. We acknowledge the observed accuracy and are committed to exploring opportunities for improving the model's performance, such as refining training strategies, increasing the dataset size, or adjusting the model architecture. We will also discuss potential limitations and future research directions in the revised manuscript to provide a more complete perspective on the system's performance capabilities.

(3) It is not clear how large was the training dataset. Also, the experimental validation is very limited/small scale and in my opinion it should be expanded to better represent functionality in real case scenario. Fig. 6 by itself is very preliminary and it is recommended to try the approach on larger scenario possibilities.

-- We thank the reviewer for this valuable comment. These points have been addressed and discussed in our revised discussion section: *"the training dataset consisted of 1452 samples, the testing dataset consisted of 161 samples."*

We apologize for not providing specific details about the training dataset in the manuscript. The size of the training dataset is a critical factor in machine learning performance. In our study, the training dataset consisted of 1452 samples. We understand that this information is essential for a comprehensive evaluation, and we will include this detail in the revised manuscript to provide greater transparency regarding the data used to train the model.

In the revised manuscript, we expand the experimental validation section to include a more diverse set of scenarios. This will provide a more comprehensive assessment of the method's robustness and effectiveness across a range of applications.

Supplementary Figure 1: Experimental confirmation of learning-based diffractive guidance in other scenes.

Supplementary Figure 1 presents results from experiments conducted in various scenes to confirm the effectiveness of the proposed model. The figure showcases nine different scenes (subfigures 1-8) with corresponding diffractive network predictions of the light field. In subfigure 1, the scene features no target of interest (ToI), while subfigures 2 to 3 include the ToI located within the $[R_4, R_5]$ region, and subfigures 4 to 8 have the ToI situated in the $[R_6, R_7]$ region. To test the model's robustness, yellow objects resembling the features of the ToI were introduced at different positions within the scenes. Additionally, the ToI is laterally displaced within the scenes, and even when it reached the edge of the scene, the proposed model demonstrated a relatively accurate prediction, highlighting its robust performance.

(4) The authors state that “The experimental validation of our proposed functional layout demonstrates its potential for rapid and intelligent capturing of real-world scenarios”. Thus, it could be good to compare the proposed performance with existing approaches. Especially to show how much processing complexity and time the proposed all-optical configuration saves in comparison to conventional digital post processing approaches.

-- We thank the reviewer for this valuable comment. To address this comment of the referee, in our revised manuscript:

Supplementary Figure 2: Performance comparison of three methods

“*” means that it can be ignored in the current measuring range. In Supplementary Figure 2, we provide a comparative analysis of the prediction time and accuracy for three different methods: the full connection neural network (FCNN), convolutional neural network (CNN), and the proposed method, with each network comprising three layers. Each training curve represents the results obtained after six rounds of training, with individual Epoch tracked for prediction accuracy variance, and average values. Notably, when the proposed optical structure is implemented, it demonstrates a significant advantage in prediction time, which is unmatched by the traditional methods.

Reviewer #2 (Remarks to the Author):

The authors describe an analog version of deep learning technique of image analysis. Comparing to its digital counterpart the analog technique offers a number of advantages well described in the paper, although requires rather lengthy fabrication of diffractive elements that will only be used for single purpose (single tracking object or a scene). The idea is new, important and timely and definitely requires careful attention. It perfectly fits the scope of the journal, would be of interest to others in the field and worth publishing.

However, the paper needs some more work. After reading it a few times and putting quite an effort to understand, I still have a few questions about the basics.

-- We sincerely thank the reviewer for his/her constructive feedback and positive evaluations.

LC element.

(1) Is it the same object (LC element) on Fig.1a with aperture about the lens size and Fig.5a with aperture about 1mm?

-- Yes. We have added more details about this question in revised manuscript: *"where the aperture of the LC layer is the same as in Fig. 1, to ensure the training data is more in line with the actual scene."* We appreciate the reviewer's attention to this detail and confirm that the object depicted in both figures is indeed the same LC element. This clarifies the continuity of the experimental setup and ensures consistency in the representation of the elements in the manuscript.

(2) Its description, lines 100-104, is unclear.

-- Thank you for pointing out the unclear description in lines 100-104. We revise the manuscript to improve the clarity and comprehensibility of this section:

"To create the desired initial alignment of liquid crystal (LC) directors, we use a configuration where a circular-patterned aluminum electrode and a planar ITO electrode are sandwiched together. The LC materials are then aligned and anchored on one smooth endface of each electrode. This setup allows us to produce modulated lightwaves that reflect the distribution of LC molecules within the LC layer."

(3) 101 "desired initial arrangement of LC directors" which is what?

-- We appreciate your observation regarding the description's lack of clarity. The desired initial arrangement of LC directors refers to the establishment of a nematic alignment of rod-shaped liquid crystal molecules within the liquid crystal layer. This initial alignment is achieved through the sandwich-like structure, creating either a nematic orientation of the liquid crystal molecules. This initial arrangement is crucial as it directly influences the response of liquid crystal molecules to external factors like electric fields, which, in turn, impacts the functionality and performance of the LC device. We provide a more detailed explanation in the revised manuscript to ensure a clearer understanding of the discussed liquid crystal molecular alignment states:

"It refers to the establishment of a nematic alignment of rod-shaped liquid crystal molecules within the liquid crystal layer. This initial alignment is achieved through the sandwich-like structure, creating either a nematic orientation of the liquid crystal molecules. This initial arrangement is crucial as it directly influences the response of liquid crystal molecules to external factors like electric fields, which, in turn, impacts the functionality and performance of the LC device."

(4) 102 "aligned and anchored" how? Planar? Homeotropic? And why?

-- Thank you for pointing out the unclear description. We revise the manuscript to improve the clarity and comprehensibility of this section:

"The alignment and anchoring of the liquid crystal (LC) molecules in this study are achieved using a planar alignment. This alignment was realized by adding an anisotropic alignment layer between the liquid crystal and the electrodes, which takes advantage of the anchoring effect on the liquid crystal's surface. The material used for the alignment layer was polyimide, and the method involved frictional rubbing of the polyimide coating with cotton cloth. Through this process, microgrooves were formed on the surface of the polyimide coating, enabling the liquid crystal's director vectors to align parallel to the direction of these grooves. Additionally, during the frictional rubbing process, the polyimide material generated polymer chains. The intermolecular forces between the liquid

crystal and the polymer chains contributed to an increased anchoring energy. This planar alignment method with the polyimide alignment layer allowed the liquid crystal molecules to align in parallel, in accordance with the predefined direction, and enabled the desired phase modulation properties and functionality of the liquid crystal layer.”

(5) 312 “In Fig. 5, the fabricated LC layers are optically characterized” and then all paragraph about fabrication, not characterisation.

-- We appreciate your feedback. To clarify and address your concern, we make the necessary revision in the manuscript as follows: *"In Fig. 5, the fabricated LC layer is visually presented,"* followed by the appropriate details of the fabrication process. This adjustment ensures that the text aligns with the content in the subsequent paragraph and accurately describes the content of Fig. 5. Thank you for bringing this to our attention.

(6) 313 “circular-patterned aluminum electrode” probably aluminium. A schematic picture would be helpful here. What are the dimensions?

-- Thank you for pointing out the potential typo and suggesting improvements. It should be *"aluminium"* electrode. A schematic diagram illustrating the circular-patterned aluminum electrode and specifying its dimensions would indeed be beneficial to provide a clearer understanding. In the revised manuscript, we include a schematic diagram of the aluminum electrode, along with its dimensions, to enhance the visual representation and comprehensibility of this element in the experimental setup. This will help provide a more detailed and visual description of the fabricated LC layers and electrodes.

Supplementary Figure 4: Schematic of circular-patterned aluminum electrode

(7) 317 “homogeneous alignment” OK. On Fig.1 it looks like homeotropic alignment.

-- We apologize for the confusion in the visual representation of the alignment in Figure 1. To reduce any potential confusion and ensure clarity, we use clearer images that reflect the homogeneous alignment mode. Additionally, in the figure caption, we explicitly highlight that the arrangement mode is **"with homogeneous alignment."**

(8) 318 "microspheres is also spread over a glass substrate" everywhere? including the LC area or only on the sides? English also needs some correction here.

-- The revised manuscript provides a more accurate description of the distribution of microspheres and ensures that the English is correctly structured. Thank you for bringing this to our attention. We make the necessary revision in the manuscript as follows: **" microspheres are also spread on the sides of the glass substrate. "**

(9) 321 Fig. 5c "Focus(mm)" do you mean focal distance or something else?

-- The label "Focus(mm)" refers to the focal distance, indicating the distance at which the LC lens achieves optimal focus. We replace "Focus(mm)" with "Focal Distance (mm)" to provide a clearer and more precise description of the plotted data.

Fig. 5 Fabrication and characterization of the functioned LC layer.

(10) As I understand the LC element is a voltage tunable lens with focal length between 4 and 8 mm. Is it right?

-- Yes. The LC element in question is indeed a voltage-tunable lens with an adjustable focal length that can vary within the range of 4.3 to 8.1 mm. This voltage-controlled tunable lens feature allows for flexibility in changing the focal length, making it suitable for a variety of optical applications. We add relevant information in manuscript as: **"The LC layer performs an adjustable focal length that can vary within the range of 4.3 to 8.1 mm."**

(11) Is the same LC lens is used for all the steps of the experiment?

-- Yes, the same LC lens is used for all the steps of the experiment. Using the same LC lens throughout the experiment is essential for maintaining the experiment's consistency, validity, and

reliability. Changing the lens during different steps could introduce variations and inconsistencies in the experimental setup, potentially leading to inaccurate results.

(12) 163 “LC layer between the primary lens and the imaging plane” if this is an additional tunable lens we need to know it’s position in the optical system.

-- Thank you for the clarification. The LC layer is an additional tunable lens in the optical system, and it is located at a distance of 8 mm from the imaging plane. We add relevant information in the manuscript as: **“which is located at a distance of 8 mm from the imaging plane”**

(13) 110 Fig. 1b Phase distributions. How were they obtained?

-- We submit our **code** and associated data in GitHub to provide transparency and a deeper understanding of our research.

<https://github.com/23Piano/A-Physics-Informed-Deep-Learning-Liquid-Crystal-Camera-with-Data-Driven-Diffractive-Guidance>

*At this stage, the code includes a partial dataset that serves the purpose of explaining the proposed method and validating its feasibility. Upon formal acceptance of the paper, we are committed to uploading the complete dataset for the benefit of the scientific community.

```
In [26]: weights = model.get_weights()
fig, ax = plt.subplots(1,3,dpi=300,figsize=(8,8))
for i in range(3):
    h = weights[i]
    if model.is_high == False:
        h = 2e-6 * tf.exp(25*h) / (tf.exp(25*h) + 1)#5*h
    plt.sca(ax[i])
    plt.xticks([])
    plt.yticks([])
    plt.imshow(h,vmin=0,vmax=2e-6,cmap=cmap1)
    # plt.colorbar(fraction=0.05)
plt.show()
```

(14) 181 “These images are employed for online optimization and assessment of the 3-layer deep diffractive phase surface.” How exactly this is done?

-- We submit our **code** and associated data in GitHub to provide transparency and a deeper understanding of our research.

<https://github.com/23Piano/A-Physics-Informed-Deep-Learning-Liquid-Crystal-Camera-with-Data-Driven-Diffractive-Guidance>

*At this stage, the code includes a partial dataset that serves the purpose of explaining the proposed method and validating its feasibility. Upon formal acceptance of the paper, we are committed to uploading the complete dataset for the benefit of the scientific community.

```

class Model(keras.Model):
    def __init__(self,
                 sampling_interval=5e-6,
                 wave_lengths=np.array([640*1e-9, 550*1e-9, 460*1e-9]),
                 is_hight = False,
                 *args, **kwargs):
        super().__init__(*args, **kwargs)
        self.sampling_interval = sampling_interval
        self.wave_lengths = wave_lengths
        self.is_hight = is_hight
        self.layer1 = DNNlayer((800, 800),self.is_hight)
        self.layer2 = DNNlayer((800, 800),self.is_hight)
        self.layer3 = DNNlayer((800, 800),self.is_hight)
        self.cross_out = CrossOut()

    def forward(self, x, distance=0.02):
        return propagate_fresnel(input_field=x,
                                distance=distance,
                                sampling_interval=self.sampling_interval,
                                wave_lengths=self.wave_lengths)

    def call(self, inputs, training=None, mask=None):
        x = tf.cast(inputs, dtype=tf.complex64)
        x = self.layer1(x)
        x = self.forward(x)
        x = self.layer2(x)
        x = self.forward(x)
        x = self.layer3(x)
        x = self.forward(x)
        if LOSS_CHOSE == 'cross':
            x = self.cross_out(x)
        return x

```

(15) Would be good to show a comparison to digital neural network arrangement. In particular, what is equivalent of the nonlinearity (the scaling function) in each layer, which is the key element of a digital neural network layer.

Supplementary Figure 2: Performance comparison of three methods

“*” means that it can be ignored in the current measuring range. In Supplementary Figure 2, we provide a comparative analysis of the prediction time and accuracy for three different methods: the full connection neural network (FCNN), convolutional neural network (CNN), and the proposed method, with each network comprising three layers. Each training curve represents the results obtained after six rounds of training, with individual Epoch tracked for prediction accuracy variance, and average values. Notably, when the proposed optical structure is implemented, it demonstrates a significant advantage in prediction time, which is unmatched by the traditional methods.

(16) 205 “several labels are assigned to each input sample respectively” how?

-- We submit our **code** and associated data in GitHub to provide transparency and a deeper understanding of our research.

<https://github.com/23Piano/A-Physics-Informed-Deep-Learning-Liquid-Crystal-Camera-with-Data-Driven-Diffractive-Guidance>

*At this stage, the code includes a partial dataset that serves the purpose of explaining the proposed method and validating its feasibility. Upon formal acceptance of the paper, we are committed to uploading the complete dataset for the benefit of the scientific community.

```
def part_mse_loss(y_true, y_pred):
    y_pred = get_intensities(y_pred)

    # r g b
    print(y_true)
    r = y_pred[:, :, :, 0] * QE[0]
    g = y_pred[:, :, :, 1] * QE[1]
    b = y_pred[:, :, :, 2] * QE[2]
    y_pred = tf.stack([r, g, b], axis=3)
    y_pred = tf.reduce_sum(y_pred, axis=3)

    # Calculate the gap between 7 areas and the targets

    ret = tf.pow(y_pred - y_true, 2)
    ret = tf.stack([
        tf.reduce_mean(
            ret[:, SI[0][0]:SI[0][1], SI[0][2]:SI[0][3]], axis=(1, 2)),
        tf.reduce_mean(
            ret[:, SI[1][0]:SI[1][1], SI[1][2]:SI[1][3]], axis=(1, 2)),
        tf.reduce_mean(
            ret[:, SI[2][0]:SI[2][1], SI[2][2]:SI[2][3]], axis=(1, 2)),
        tf.reduce_mean(
            ret[:, SI[3][0]:SI[3][1], SI[3][2]:SI[3][3]], axis=(1, 2)),
        tf.reduce_mean(
            ret[:, SI[4][0]:SI[4][1], SI[4][2]:SI[4][3]], axis=(1, 2)),
        tf.reduce_mean(
            ret[:, SI[5][0]:SI[5][1], SI[5][2]:SI[5][3]], axis=(1, 2)),
        tf.reduce_mean(
            ret[:, SI[6][0]:SI[6][1], SI[6][2]:SI[6][3]], axis=(1, 2))
    ], 0)

    return tf.reduce_mean(ret)
```

(17) 225 “The first column is the image...” “columns 1-3 are numerical results” probably columns 2-4?

-- We appreciate your correction. It should be **"2-4 are numerical results"**. Thank you for pointing out the error, and we acknowledge and appreciate your attention to detail. The mistake will be corrected in the revised manuscript to accurately reflect the content and presentation of the data.

(18) 233 “typical LC molecule distributing pattern across various LC molecule arrangement modes.” Unclear what it is and how this is used.

-- "typical LC molecule distributing pattern across various LC molecule arrangement modes" refers to different patterns in which the liquid crystal (LC) molecules are distributed within the LC layer. Each of these patterns corresponds to a specific working focal length or target capture depth within the LC layer. The optical system employs these various distribution modes to selectively achieve

different working focal lengths or depths of field, enabling the tracking of Targets of Interest (ToI). We give a clearer explanation in the revised manuscript as:

“which refers to different patterns in which the liquid crystal (LC) molecules are distributed within the LC layer. Each of these patterns corresponds to a specific working focal length or target capture depth within the LC layer. The proposed optical system employs these various distribution modes to selectively achieve different working focal lengths or depths of field, enabling the tracking of ToI.”

Reviewer #3 (Remarks to the Author):

In this paper, the authors presented an LCD camera with diffractive surface-based guidance to track a target in the depth direction. They employed a deep diffractive system to predict the LCD voltage, subsequently adjusting the PSF/focal plane of the system. However, the manuscript lacks some pivotal convincing results and I cannot recommend it for publication.

-- We sincerely thank the reviewer for his/her constructive feedback. The concerns raised regarding the manuscript's lack of pivotal convincing results are indeed important and warrant careful consideration. To address these concerns, we make the following revisions to the manuscript:

1. **Enhanced Experimental Validation:** We expand the experimental validation section to provide a more comprehensive assessment of the proposed camera system's performance, including a broader range of scenarios and datasets. This will include quantifying system performance in terms of tracking accuracy, robustness, and efficiency in the depth direction.

Supplementary Figure 1: Experimental confirmation of learning-based diffractive guidance in other scenes.

2. **Comparison with Existing Approaches:** We conduct a comparative analysis to demonstrate how the proposed LC camera system performs in comparison to existing techniques (conventional digital post processing approaches), highlighting its advantages in terms of tracking capabilities and depth direction guidance.

Supplementary Figure 2: Performance comparison of three methods

3. **Code Available:** By making our code and data available, we aim to enhance the credibility and replicability of our work. We understand the importance of robust results in scientific research, and we are dedicated to providing the necessary resources for the scientific community to assess and replicate our findings. This transparency will contribute to a more convincing and rigorous evaluation of our work.

<https://github.com/23Piano/A-Physics-Informed-Deep-Learning-Liquid-Crystal-Camera-with-Data-Driven-Diffractive-Guidance>

*At this stage, the code includes the partial data that serves the purpose of explaining the proposed method and validating its feasibility. Upon formal acceptance of the paper, we are committed to uploading the complete dataset for the benefit of the scientific community.

(1) The author showed the proposed architecture in Fig. 1, yet it is confusing. From Fig. 1a, it seems that the authors utilize diffractive phase surfaces to predict the optimal reorientation of the LC molecules, and tracked the target from the complex background using LC molecules only. Conversely, Figure 1c suggests that the LC layers are integrated with the diffractive surfaces,

implying that both the LC layers and diffractive surfaces collaboratively track the target. The authors must elucidate their methodology more distinctly.

-- Thank you for pointing out the unclear description. Figure 1c illustrates that the LC layer and diffractive surfaces are physically integrated and closely coupled. This visual representation is intended to show the practical arrangement of these components.

It is essential to clarify that the entire optical computational process is achieved through the diffractive phase surfaces. The LC layer plays a role as they are modulated by electrical signals to adjust the wavefront, which is an essential part of the adaptive optics process. So, while the LC layers and diffractive surfaces are closely integrated, the primary optical computation and the prediction of optimal reorientation are accomplished through the diffractive phase surfaces. The LC layers act as components that are electrically controlled to fine-tune the wavefront. In the revised manuscript, we will provide a more distinct and detailed explanation of how these elements work together to achieve the desired optical outcomes, clarifying the roles of each component. This will ensure that readers have a clearer understanding of the methodology.

It is important to emphasize that, although the LC layers and diffractive surfaces are integrated, the primary optical computation and the forecasting of optimal reorientation are achieved via the diffractive phase surfaces. The LC layers serve as integral components subject to precise electrical control for the refinement of the wavefront.

(2) Following the previous question, the authors mentioned a primary lens was employed. Is it used in “optical inference” or “ToI tracking from complex background”?

-- We appreciate your feedback. We give a clearer explanation in the revised manuscript as: *“which is used for lightwave compression and imaging in ToI tracking from complex background.”*

(3) In the paper, the authors stated that they “only performed spatial segmentation in the depth direction to track the ToIs region”. What’s the axial resolution achieved by the deep diffractive voting process? A characterization of how performance is impacted by the position of ToI within the FOV (on x-y plane) is needed.

-- Thank you for your question. The axial resolution achieved by the deep diffractive voting process is influenced by several factors, including the distance between the LC layer and the CMOS plane and the parameters of the primary lens. In the current architecture, the axial resolution is approximately 8.4 cm.

In terms of the impact of the ToI's position within the FOV on x-y plane, a detailed characterization is shown in Fig. 1 to assess how performance varies with different positions. This characterization can help in understanding the system's capabilities for tracking targets at different depths and locations within the field of view.

Supplementary Figure 1 presents results from experiments conducted in various scenes to confirm the effectiveness of the proposed model. The figure showcases nine different scenes (subfigures 1-8) with corresponding diffractive network predictions of the light field. In subfigure 1, the scene features no target of interest (ToI), while subfigures 2 to 2 include the ToI located within the $[R_4, R_5]$ region, and subfigures 4 to 8 have the ToI situated in the $[R_6, R_7]$ region. To test the model's robustness, yellow objects resembling the features of the ToI were introduced at different positions within the scenes. Additionally, the ToI is laterally displaced within the scenes, and even when it reached the edge of the scene, the proposed model demonstrated a relatively accurate prediction, highlighting its robust performance.

Supplementary Figure 1: Experimental confirmation of learning-based diffractive guidance in other scenes.

(4) Regarding each signal voltage level, is it a singular value or an electric field distribution across the LCD layer? If it's a distribution, how do the authors determine it?

-- Thank you for pointing out the unclear description. In our study, the signal voltage levels correspond to the electric field distributions across the LC layer. To achieve this distribution, we have designed a sandwich structure for the LC layer, along with patterned electrodes. By adjusting

the voltage applied to the electrodes, we can effectively switch and control the electric field distribution within the LC layer. The specific details of the electric field distribution switching are visually represented in the Supplementary Figure 3.

Supplementary Figure 3: Mode switching of the electric potential distributions across the LC layer

Supplementary Figure 3 presents a comprehensive illustration of the mode switching characteristics within the LC layer of the 1mm aperture-patterned electrode configuration. The simulation captures the modal responses of the LC layer under varying voltage conditions, namely 5V, 20V, and 30V. The corresponding electric potential distributions are visualized, providing insight into how the electric field is modulated within the LC layer. Additionally, the figure showcases the distribution of LC molecule orientation vectors, demonstrating the dynamic response of the LC layer to different voltage conditions.

(5) Throughout the manuscript, the authors emphasize tracking the target of interest. However, both the simulation and experimental results primarily depict the prediction of LC molecule arrangements via deep diffractive layers. It would be beneficial for the authors to provide visual evidence of the actual target tracking.

-- We understand the reviewer's concern regarding providing visual evidence of actual target tracking, which is an important aspect of our proposed methodology. To address this, we have taken several steps:

Experimental Optical Prediction Videos: We have included the experimental optical prediction videos to visually demonstrate the feasibility of our proposed method. These videos help provide clear visual evidence of the prediction process and how it relates to target tracking.

Supplementary Video

Open Source Code: In addition to the videos, we have made our code available to readers. This code allows for a deeper understanding of our entire architecture and facilitates reproducibility. It provides a practical means to comprehend how the optical calculations are performed, including target tracking.

<https://github.com/23Piano/A-Physics-Informed-Deep-Learning-Liquid-Crystal-Camera-with-Data-Driven-Diffractive-Guidance>

*At this stage, the code includes a partial dataset that serves the purpose of explaining the proposed method and validating its feasibility. Upon formal acceptance of the paper, we are committed to uploading the complete dataset for the benefit of the scientific community.

Core Methodology: It's important to note that the core of our methodology is centered around spatial localization of the target through optical computations in tracking the target of interest process. We have aimed to comprehensively convey this aspect throughout the manuscript, as it is fundamental to the proposed approach.

We have strived to present a well-rounded perspective on our approach, including both theoretical and practical aspects. In the revised manuscript, we will continue to emphasize these points and provide clear references to the videos and code to support the understanding and validation of our proposed method.

(6) I observed marked differences between the input scenes in Fig. 3a and Fig. 6a. The field of view (FOV) in Figure 3a appears more constricted, with objects seeming blurry. In contrast, the fov in Figure 6a is considerably broader, presenting objects with clarity. Could the authors clarify the reasons behind this discrepancy?

--Thank you for raising this observation. To address the difference between the input scenes in Figure 3a and Figure 6a:

The key element of this method involves training the optical network to recognize the degree of blurriness of the target of interest in relation to the point spread function (PSF) of the primary lens. This recognition is essential for spatially confirming the position of the target. However, it's important to note that there are reasons behind the apparent differences between these figures:

Blurry Differences: The primary reason for the visual contrast between Figure 3a and Figure 6a is that they represent different stages of the experimentation. Figure 3a corresponds to the simulated input, where we can control parameters for illustrative purposes. In contrast, Figure 6a represents the actual experimental input. During experiments, it is often necessary to introduce a stronger background illumination to enhance the contrast of the optical network's predicted light field. This can lead to a perceived difference in the visual representation. *In the additional figure, we adjust the intensity and angle of the background illumination to weaken this illusion.*

Field of View (FOV) Differences: The varying field of view is due to the selection of a larger FOV configuration in the experimental setup (Figure 6a) to provide a clearer representation of the real-world experimental scenario. We have slightly corrected the position of LC layer to cope with the actual experiments. This change in FOV is made to better demonstrate the practical application and does not fundamentally alter the verification of the method's feasibility.

In summary, the discrepancy in appearance between the simulated and experimental inputs is primarily due to variations in lighting conditions and the choice of field of view for illustrative and clarity purposes. It does not impact the fundamental validity or feasibility of the method.

(7) “1-3 are numerical results ...” in line 226 should be “2-4 are numerical results...”

-- We appreciate your correction. It should be *" 2-4 are numerical results "*. Thank you for pointing out the error, and we acknowledge and appreciate your attention to detail. The mistake will be corrected in the revised manuscript to accurately reflect the content and presentation of the data.

Reviewers' comments:

Reviewer #1 (Remarks to the Author):

The revised paper addresses most of my comments and I think it can now be accepted for publication. However, I still think the authors should address one more point to make the manuscript more comprehensive. The authors demonstrated some tunability in changing the target of interest but in my opinion, it is still not clear what are the limits that could be obtained in the proposed configuration. How tunable the z-stack approach exactly is? This should be demonstrated or at least explained in a more quantitative manner.

Reviewer #2 (Remarks to the Author):

The authors responded to all of my (and the other reviewers) notes, made the corrections and added the supplementary material to explain some details. The explanations in the rebuttal letter and in the supplementary material make the paper more understandable now. However, in the paper there is no references to the supplementary materials and the rebuttal letter is not visible to a reader. I believe that most of the explanations has to be in the paper text.

Reviewer #3 (Remarks to the Author):

I think the authors have done a reasonable job in revising their manuscript in response to most of the referee comments. It reads better now.

Reviewer #1 (Remarks to the Author):

The revised paper addresses most of my comments and I think it can now be accepted for publication. However, I still think the authors should address one more point to make the manuscript more comprehensive. The authors demonstrated some tunability in changing the target of interest but in my opinion, it is still not clear what are the limits that could be obtained in the proposed configuration. How tunable the z-stack approach exactly is? This should be demonstrated or at least explained in a more quantitative manner.

-- We appreciate the time and effort you have dedicated to evaluating our manuscript. To address this comment, we have provided a clearer understanding of the limits. We have decided to remove the term "z-stack" from the manuscript because we believe its inclusion could potentially lead to misinterpretation among readers. The physical structure of the utilized optical neural network can be easily understood from Figure 1C.

“Achieving optimal results and pushing the boundaries of this configuration will require advancements in both manufacturing accuracy and the sophistication of optical propagation models.”

“... the accuracy achieved under specific conditions and may not necessarily reflect the performance limit of the proposed configuration. The achievable performance can be influenced by various factors, including the complexity of the ToIs, the quality of the training dataset, and the errors in physical processing.”

Reviewer #2 (Remarks to the Author):

The authors responded to all of my (and the other reviewers) notes, made the corrections and added the supplementary material to explain some details. The explanations in the rebuttal letter and in the supplementary material make the paper more understandable now. However, in the paper there is no references to the supplementary materials and the rebuttal letter is nor visible to a reader. I believe that most of the explanations has to be in the paper text.

-- We appreciate the time and effort you have dedicated to evaluating our manuscript. We have made the required revisions to integrate essential explanations and content directly into the paper. We also hope to make public the review comments and responses

“The primary reason for the visual contrast between scenes in Figure 3A and Figure 6A is that they represent different stages of the experimentation. Figure 3A corresponds to the numerical input, where we can control parameters for illustrative purposes. In contrast, Figure 6A represents the actual experimental input. During experiments, it is often necessary to introduce a stronger background illumination to enhance the contrast of the optical network's predicted light field. This can lead to a perceived difference in the visual representation. In the Supplementary Figure 1, we adjust the intensity and angle of the background illumination to weaken this illusion.”

“We also provide a comparative analysis of the prediction time and accuracy for proposed method and conventional digital neural network in Supplementary Figure 2”

“In our study, the signal voltage levels correspond to the electric field distributions across the LC layer. To achieve this distribution, we have designed a sandwich structure for the LC layer, along with patterned electrodes. By adjusting the voltage applied to the electrodes, we can effectively switch and control the electric field distribution within the LC layer. The specific details of the electric field distribution switching are visually represented in the Supplementary Figure 3.”

In the Supplementary Figure 4, we include a schematic diagram of the aluminum electrode, along with its dimensions, to enhance the visual representation and comprehensibility of this element in the experimental setup.

Reviewer #3 (Remarks to the Author):

I think the authors have done a reasonable job in revising their manuscript in response to most of the referee comments. It reads better now.

-- We appreciate the time and effort you have dedicated to evaluating our manuscript.

REVIEWERS' COMMENTS:

Reviewer #1 (Remarks to the Author):

The authors addressed my concerns and I now recommend accepting the revised manuscript for publication.

Reviewer #1 (Remarks to the Author):

The authors addressed my concerns and I now recommend accepting the revised manuscript for publication.

-- Thank you for your time and dedication in reviewing our manuscript.